# Decreased Prosaposin and Progranulin in the Cingulate Cortex Are Associated with Schizophrenia Pathophysiology

**DOI:** 10.3390/ijms231912056

**Published:** 2022-10-10

**Authors:** Yachao He, Xiaoqun Zhang, Ivana Flais, Per Svenningsson

**Affiliations:** 1Department of Clinical Neuroscience, Karolinska Institutet, 17177 Stockholm, Sweden; 2CNS Department, Boehringer Ingelheim Pharma GmbH & Co. KG, 55216 Biberach, Germany; 3Department of Basic and Clinical Neuroscience, Institute of Psychiatry, Psychology and Neuroscience, King’s College London, London WC2R 2LS, UK

**Keywords:** prosaposin, progranulin, schizophrenia, sphingolipids, cingulate cortex

## Abstract

Prosaposin (PSAP) and progranulin (PGRN) are two lysosomal proteins that interact and modulate the metabolism of lipids, particularly sphingolipids. Alterations in sphingolipid metabolism have been found in schizophrenia. Genetic associations of PSAP and PGRN with schizophrenia have been reported. To further clarify the role of PSAP and PGRN in schizophrenia, we examined PSAP and PGRN levels in postmortem cingulate cortex tissue from healthy controls along with patients who had suffered from schizophrenia, bipolar disorder, or major depressive disorder. We found that PSAP and PGRN levels are reduced specifically in schizophrenia patients. To understand the role of PSAP in the cingulate cortex, we used an AAV strategy to knock down PSAP in neurons located in this region. Neuronal PSAP knockdown led to the downregulation of neuronal PGRN levels and behavioral abnormalities. Cingulate-PSAP-deficient mice exhibited increased anxiety-like behavior and impaired prepulse inhibition, as well as intact locomotion, working memory, and a depression-like state. The behavioral changes were accompanied by increased early growth response protein 1 (EGR-1) and activity-dependent cytoskeleton-associated protein (ARC) levels in the sensorimotor cortex and hippocampus, regions implicated in circuitry dysfunction in schizophrenia. In conclusion, PSAP and PGRN downregulation in the cingulate cortex is associated with schizophrenia pathophysiology.

## 1. Introduction

Schizophrenia is a complex chronic psychiatric disorder involving neurodevelopmental malfunction [1]. Core manifestations of schizophrenia can be categorized into positive (delusions, hallucinations, and disorganized behaviors), negative (social withdrawal, diminished motivation, apathy), and cognitive symptoms [1]. A complex interplay between genetic and environmental factors is suggested to underlie the development of schizophrenia [2]. A variety of brain regions exhibit functional or structural changes contributing to different symptoms [2]. The anterior cingulate cortex shows a volume decrease in early phase of the disease [3], and the perigenual cingulate cortex has been identified as both structurally and functionally abnormal in schizophrenia [4]. However, the molecular mechanisms involved in the pathogenesis of schizophrenia remain elusive. Oxidative stress and inflammatory processes have been pinpointed as key mechanisms that dysregulate the development and maintenance of synaptic function, as well as myelination, resulting in schizophrenia [5].

Prosaposin (PSAP) is the precursor protein of four lysosomal sphingolipid hydrolase activator proteins known as saposins A–D [6]. Full-length PSAP is recognized as a secreted neurotrophic factor [7]. Recently, PSAP has been shown to specifically sensitize neurons, rather than glial cells, against oxidative stress when deleted [8]. PSAP deficiency causes lysosomal storage disorders [9]. PSAP mutations are linked to Parkinson’s disease (PD) [10], and an association of single nucleotide polymorphisms (SNPs) in the PSAP gene with schizophrenia was revealed in a Dutch population [11]. Meanwhile, as the main substrate of saposins, sphingolipid metabolism is abnormal in patients with schizophrenia [12].

Progranulin (PGRN) shares similar identities of lysosomal protein and neurotrophic factor as PSAP [13]. Mutations in GRN, the gene that encodes PGRN, cause familial frontotemporal lobar dementia (FTLD) and Lewy body dementia (LBD) [14,15]. Several drug development efforts in FTLD target PGRN [16]. PGRN and PSAP form a dimer and facilitate each other’s trafficking both intracellularly and extracellularly [13]. It has been reported that PGRN regulates the activity of enzymes related to sphingolipid metabolism directly or by modulating PSAP trafficking in the brain [17,18,19]. Intriguingly, it has been shown that schizophrenia patients exhibit tau and TAR DNA binding protein-43 (TDP-43) deposits in the brain, which is typical of FTLD brains [20,21]. Furthermore, loss-of-function GRN mutation was found in two siblings diagnosed as schizophrenia and FTD, respectively [22], and GRN SNPs have been linked to risk of schizophrenia [23].

Given these premises, we examined the protein levels of PSAP and PGRN in postmortem cingulate cortex tissue from healthy controls and schizophrenia patients, using bipolar disorder and major depressive disorder as disease controls. To study the in vivo effects of PSAP and PGRN, we knocked down PSAP in the cingulate cortex of PSAP gene-floxed mice by adeno-associated-virus (AAV)-mediated overexpression of Cre recombinase. We studied these cingulate-PSAP-deficient mice in a battery of behavior tests, and with immediate early gene mapping of the forebrain.

## 2. Results

### 2.1. PSAP and PGRN Are Decreased in the Cingulate Cortex of Schizophrenia Patients

Apart from the connections with neurodegenerative diseases, including PD, FTLD, and LBD, PSAP and PGRN are also genetically linked to schizophrenia [11,22,23]. However, there is still a lack of investigation into their protein levels in brain tissue, especially the cingulate cortex, in schizophrenia patients. As bipolar disorder and major depressive disorder share common etiological mechanisms with schizophrenia [24], we also included brain tissue from patients with these two diseases. PSAP, PGRN, and β-actin protein levels were quantified by Western blotting analysis in postmortem cingulate cortex tissue from healthy controls and patients suffering from schizophrenia, bipolar disorder, or major depressive disorder (Figure 1A). Interestingly, significantly decreased PSAP levels were found exclusively in schizophrenia patients compared with healthy controls, but not in patients with bipolar disorder or major depressive disorder (Figure 1B). Similarly, a significant decrease in PGRN levels was also found in schizophrenia patients, and there were also decreasing trends in the other two groups (Figure 1C). No association was found when PSAP and PGRN levels were correlated with lifetime quantity of fluphenazine or equivalent neuroleptic in schizophrenia and bipolar disorder patients (Figure 1D,G).

### 2.2. Deletion of PSAP in the Cingulate Cortex Causes Downregulation of PGRN Levels

Interdependency has been revealed between PSAP and PGRN [13]. Since we found decreases in both proteins in the cingulate cortex of schizophrenia patients, and PSAP changes seemed more specific than PGRN, we wondered whether PSAP deletion in the cingulate cortex of mice will influence the levels of PGRN and mouse behavior. Here we utilized a mouse line in which the PSAP gene was floxed. Two groups of PSAP-floxed mice received AAV-GFP or AAV-GFP-Cre bilateral injections, respectively, in the cingulate cortex, at two sites in each hemisphere (Figure 2A). Genetic knockdown of PSAP in the cingulate cortex of mice that received AAV-GFP-Cre injection was confirmed by fluorescent in situ hybridization (RNAScope) (Figure 2A). Furthermore, immunofluorescent staining of PSAP, PGRN, Cre, and GFP showed a co-expression of PSAP and PGRN in cells and specific knockdown of PSAP in Cre-expressing neurons (Figure 2B). Mean fluorescence intensity quantification of PSAP staining in GFP- and Cre-expressing neurons demonstrated a clear loss of PSAP in the cingulate cortex of AAV-GFP-Cre-injected mice compared with AAV-GFP-injected mice (Figure 2B,C). Furthermore, PSAP knockdown caused the downregulation of PGRN in the same neurons (Figure 2B,D). 

### 2.3. Mice with PSAP Deficiency in the Cingulate Cortex Display Anxiety-Like Behavior and Sensorimotor Gating Dysfunction

To investigate the influence of PSAP deletion in the cingulate cortex on behavior, tests examining locomotion, emotion, cognition, and sensorimotor gating were performed on mice. Eight weeks after AAV injection, mice were subjected to open field tests. The locomotion of the mice was not altered by PSAP loss in the cingulate cortex, as shown by the distance traveled in the open field (Figure 3A,B). To evaluate the depressive-like state of the mice, a forced swim test was conducted, and no difference was found between the two groups of mice (Figure 3C). Furthermore, to assess the working memory of the mice, a Y maze test was carried out, and the performances of the two groups of mice did not differ (Figure 3D). However, in the light–dark transition test, AAV-GFP-Cre-injected mice showed elevated anxiety-like behavior compared with AAV-GFP-injected mice, as demonstrated both by the distance traveled (Figure 3E,F) and time spent in the light box (Figure 3E,G) by the mice. In the PPI test, mice injected with AAV-GFP-Cre showed impaired sensorimotor gating function, as revealed by the PPI magnitude detected with a prepulse intensity of 12 dB above background (Figure 3H).

### 2.4. PSAP Deficiency in the Cingulate Cortex Elevates EGR-1 and ARC Levels in Different Brain Regions

Immediate early genes are commonly used to map alterations in brain circuitries [25]. Early growth response protein 1 (EGR-1) and activity-dependent cytoskeleton-associated protein (ARC) are two immediate early genes associated with schizophrenia [26,27,28,29]. To investigate the effect of PSAP deletion in the cingulate cortex on EGR-1 and ARC mRNA levels, we performed radioactive in situ hybridization to measure EGR-1 and ARC mRNA levels on brain sections of these mice. Interestingly, PSAP deletion in the cingulate cortex caused a widespread upregulation of EGR-1, especially in the sensorimotor cortex, and hippocampus (Figure 4A,B). Meanwhile, an increase in ARC level was detected in the hippocampus, but not in other studied regions (Figure 4A,C).

## 3. Discussion

PSAP and PGRN have been recognized as playing a vital role in neurodegenerative diseases. Variants in the saposin D domain of the PSAP gene cause autosomal dominant PD [10], and GRN haploinsufficiency results in FTLD [14]. Deleterious GRN mutations have also been revealed to be a rare cause of familial LBD [15]. Interestingly, SNPs in PSAP and PGRN were found to be associated with schizophrenia [11,22,23]. Although schizophrenia and PD are characterized by opposite alterations in the dopamine system of the central nervous system (CNS), these two diseases share genetic causes [30]. Moreover, a recent study has reported that the risk of contracting PD can be increased by schizophrenia spectrum disorders [31]. Vice versa, another study showed that increased risk of schizophrenia is associated with increased genetic risk of PD [32]. Likewise, schizophrenia shows a similar cognitive decline or behavioral changes as displayed by FTLD, especially behavioral variant frontotemporal dementia (bv-FTD), and LBD. It has been reported that the risk of schizophrenia in relatives of FTD probands is much higher than in relatives of AD probands [33]. Therefore, the involvement of PSAP and PGRN in schizophrenia pathogenesis is possible. Since PSAP and PGRN are emerging targets for therapeutic development in FTLD, and therapies against negative symptoms are lacking in schizophrenia, it is highly relevant to study PSAP and PGRN levels in schizophrenia. Consistent with the genetic studies, we report that PSAP and PGRN are decreased in the cingulate cortex of patients with schizophrenia, but not in patients with bipolar disorder or depression. The PSAP or PGRN levels are not influenced by neuroleptic treatment; their decreases are related to schizophrenia pathogenesis.

The interaction of PSAP and PGRN has been shown to dimerize and modulate their levels and activities [13]. Accordingly, we found that genetic knockdown of neuronal PSAP in the cingulate cortex of mice led to a decrease in PGRN in the same neurons. To investigate the influence of neuronal PSAP deletion on mouse behavior, a series of behavioral tests were conducted. Interestingly, cingulate-PSAP-deficient mice presented normal locomotion, a non-depressive state, and an intact working memory, but suffered from increased anxiety, and, more importantly, impaired sensorimotor gating. However, PSAP and PGRN are also expressed in glial cells, especially microglia [34,35]. Considering their trafficking between cells, the behavioral influence of neuronal PSAP removal could be affected by glial-cell-originated PSAP. Anxiety is a common co-morbidity of schizophrenia, and has been linked to changes in the cingulate cortex, including altered proteomes, metabolic profiles, and mitochondrial pathways [36,37]. Functionally, activity changes in the cingulate cortex have also been reported in anxiety disorder [38,39]. We found a reduction in PPI in the cingulate-PSAP-deficient mice. Our data agree with the existing literature on a critical role of the cingulate cortex in sensorimotor gating in rodents [40,41,42]. Indeed, a previous study has shown that cytotoxic medial prefrontal cortex lesions, which damage such regions as the dorsal anterior cingulate areas, attenuate the PPI response [40]. Moreover, decreased medial prefrontal cortex (including the cingulate cortex) activity and volume are also associated with reduced PPI [41]. Furthermore, the hyperactive anterior cingulate cortex–mediodorsal thalamus pathway has been reported to suppress PPI [42]. As the cingulate cortex is well known as an important interface regarding emotion, action, and memory [43], it is not surprising that PSAP deficiency in the cingulate cortex caused a widespread influence on immediate early genes in the brain. In this context, it is noteworthy that several lines of evidence point towards a role of EGR-1 and ARC in schizophrenia [26,27,28,29], and that was why we decided to focus our studies on these two immediate early genes.

At a cellular level, both PSAP and PGRN are indispensable factors for the metabolism of sphingolipids. Saposin A optimizes the hydrolysis of galactocerebrosides; saposin B facilitates the hydrolysis of galactocerebrosides sulfate, globotriaosylceramide, GM1 ganglioside, and glycerolipid; saposin C mainly enhances the hydrolysis of glucocerebroside; saposin D participates in the hydrolysis of sphingomyelin and ceramide [9]. Regarding PGRN, a study has shown that recombinant PGRN increases the expression and lysosomal delivery of hexosaminidase A and alleviates GM2 ganglioside accumulation [17]. Additionally, PGRN directly interacts with glucocerebrosidase and affects its processing and trafficking to the lysosome, and indirectly modulates glucocerebrosidase activity by facilitating PSAP processing [18]. A recent lipidomic study revealed dramatic changes in sphingolipids and bis(monoacylglycero)phosphate (BMP) in PGRN-deficient mouse brains [44]. Accumulating evidence has highlighted the role of sphingolipids in schizophrenia. Expression alterations of genes related to sphingolipid metabolism have been found in the prefrontal cortex of subjects with schizophrenia [45]. Sphingomyelin and galactocerebrosides 1 and 2 were reported to be reduced in the thalami of schizophrenic patients [46]. Elevated ceramide levels were revealed specifically in white matter instead of grey matter in schizophrenic patients [47]. Lower sphingosine-1-phosphate levels were observed in the corpus callosum of patients with schizophrenia compared to patients with major depressive disorder or bipolar disorder [48]. In the cerebrospinal fluid (CSF), sphingolipids are associated with clinical scores in first-episode psychosis patients [49]. Peripherally, aberrant alterations in sphingomyelin and ceramide levels in the red blood cells and skin are strongly linked to schizophrenia [50]. It has even been hypothesized that the disruption of the acid sphingomyelinase/ceramide system contributes to the dopamine neurotransmission malfunction, neuroinflammation, and oxidative stress, which are the main theories regarding the cause of schizophrenia [12]. Interestingly, PSAP plays a fundamental role in the sensitization of neurons to oxidative stress [8], and PGRN is also a neuroinflammation modulator [35]. Moreover, it has been shown that PGRN exerts inflammatory effects through NF-κB pathway [51,52], and recent studies have implicated this pathway in schizophrenia [53,54]. Therefore, future mechanistic studies on the role of PSAP and PGRN in schizophrenia could bring novel insights into the pathogenesis of schizophrenia, with implications for drug development.

One limitation of the study is that the sample size of human brain tissue for the expression analysis is relatively small, which may have caused false negative results of PGRN changes in BP and MDD groups. Another limitation is that the studies on EGR-1 and ARC were conducted 30 min after FST, but not PPI, which in retrospect would have made more sense. Although the 30 min gap has been used for EGR-1 and ARC studies [55,56], there are studies reporting that the induction peak of ARC mRNA occurs in about 5–10 min [57,58], while EGR-1 protein levels reach maximum after 1 h [59]. Therefore, a study focused on the regulation of EGR-1 and ARC at several time points after PPI testing in cingulate-PSAP-deficient mice would be informative.

In conclusion, our study extends to previous human genetic findings, and adds neuropathological and in vivo animal data, suggesting an important role of PSAP and PGRN in schizophrenia pathophysiology.

## 4. Materials and Methods

### 4.1. Human Brain Samples

#### Human Brain Samples

All human brain samples were obtained from the well-established brain bank at the Stanley Neuropathology Consortium [60]. All samples were stored at −80 °C until use. Informed consent forms were signed by all donors for the use of their brain tissue in this research. The work with postmortem human brain tissue has been approved by the Ethics Committee at Karolinska Institute (reg. no. 2014/1366-31). Demographics and clinical information of the participants are shown in Table 1.

### 4.2. Western Blotting

Frozen cingulate cortex tissue blocks were dissected on a platform cooled by dry ice. Around 100 mg of tissue was obtained from each sample and homogenized with a sonicator in RIPA buffer with protease inhibitor cocktail tablet (#A32963, Thermo Fisher Scientific, Waltham, MA, USA) and phosphatase inhibitor (PhosSTOP, #4906837001, Merck, NJ, USA). After homogenization, the lysate was incubated with agitation for 30 min at 4 °C, followed by centrifugation at 14,000 rpm for 20 min at 4 °C. Upon centrifugation, the supernatant was stored in an Eppendorf tube, while the pellet on the bottom was discarded. Concentrations of all samples were measured with the BCA Protein Assay Kit (#23225, Thermo Fisher Scientific, MA, USA) according to the protocol provided by the manufacturer. All samples were stored at −80 °C until use. For Western blotting, samples were mixed with 4× Laemmli protein sample buffer (#1610747, Bio-Rad, CA, USA) and denatured by heating for 10 min at 95 °C. Ten microliters of each sample was loaded in precast 12% Mini-PROTEAN TGX Stain-Free Protein Gels (#4568046, Bio-Rad, CA, USA) and resolved by electrophoresis in a Bio-Rad mini-PROTEAN System. Protein was then transferred to a 0.2 μm nitrocellulose membrane with a Trans-Blot Turbo Transfer System (Bio-Rad). After blocking of unspecific binding sites with blocking buffer (#927-60001, LI-COR, NE, USA), membranes were incubated with primary antibodies, including rabbit anti-PSAP (1:1000, #HPA004426, Atlas Antibodies, Bromma, Sweden), rabbit anti-Granulin (1:1000, #ab187070, Abcam, Cambs, UK), mouse anti-β-actin (1:6000, #A5441-100UL, Sigma-Aldrich, MO, USA). Primary antibodies were detected using fluorophore-conjugated goat anti-rabbit (1:10000, IRDye 680CW) or goat anti-mouse (1:10,000, IRDye 800CW) secondary antibody, and scanned on an LI-COR Imaging System (Odyssey DLx, LI-COR, NE, USA). All bands were quantified with Image Studio (LI-COR, NE, USA). All uncropped original blots are provided in Appendix A.

### 4.3. Stereotaxic Surgery on Mice

All animal experiments were approved by the local Animal Ethics Committee (5018-2018). Conditional PSAP gene-targeted mice (C57BL/6J) were generated by Cyagen (Santa Clara, CA, USA) by flanking exon 2–4 of the PSAP gene with loxp sites (Floxed). Primers used for genotyping were: F1 (GCAGAAGATGCAGGACCGTGTG) and R1 (ATCACTGGGTCTCCCTAGCCCAAG), F2 (ATTCCTGACTCCTGCATCCTG) and R2 (CACCTTCCTCACAAACCCCTG). Mice were kept in rooms with 12-h light/dark cycles and controlled temperature/humidity (20 °C/53%), and were provided with food pellets and water on an ad libitum basis. Five- to eight-month-old mice were used in this study. Mice in two groups were age and gender matched. All mice underwent surgical procedures under general anesthesia induced by isoflurane (3% for induction, 1% for maintenance) with 0.5 lpm air flow. Mice were mounted on a stereotaxic frame (Kopf) with their heads fixed and aligned horizontally and vertically to the frame. Thermal support was provided. Mouse eyes were protected from drying out during surgery by eye lubricant. Local skin anesthesia was achieved by application of lidocaine. Coordinates for four sites in the cingulate cortex used for injection were 1/2 (AP = 1.0 mm, ML = ±0.4 mm, DV = −1.8 mm) and 3/4 (AP = 0.2 mm, ML = ±0.4 m, DV = −1.8 mm), as calculated relative to bregma according to the stereotaxic atlas (Paxinos and Franklin, 2001). Hamilton syringes (5 μL) with a 33 G needle were used for AAV injections. For each site, one microliter of AAV solution was injected at a rate of 0.2 μL/min. The titer both for AAV-GFP and AAV-GFP-Cre was 1 × 1012 gc/mL. Upon injection, the syringe was kept in the position for 5 min and slowly retracted afterwards. Incisions were stitched and cleaned. Temgesic (i.p.) treatment was given to mice once per day for three consecutive days. A synapsin promoter was used in both AAV-GFP-Cre and AAV-GFP.

### 4.4. Fluorescent In Situ Hybridization (FISH)

FISH was performed with an RNAScope Fluorescent Multiplex Assay (Advanced Cell Diagnostics, CA, USA). Mouse brains were perfused with 1 × PBS, followed by 4% paraformaldehyde in 1 × PBS solution. Dissected out mouse brains were further post-fixed in 4% paraformaldehyde in 1 × PBS solution for 24 h, then dehydrated in 30% sucrose until tissue sank. Processed brains were embedded in OCT. A cryostat (Leica CM 3050S, Wetzlar, Germany) was used for brain slicing. Free-floating sections were kept in antifreeze solution stored in a −20 °C freezer until experiment. On the experiment day, free-floating sections were washed in 1 × PBS and mounted on a Superfrost slide and dried out in room temperature for 2 h. After washing in 1 × PBS, sections were boiled in Pretreat II for target retrieval for 5 min. Then, sections were washed in DEPC water and dehydrated in 100% ethanol, followed by incubation with protease III for 30 min at 40 °C. Upon washing, sections were hybridized with PSAP probe (Mm-Psap-O1, cat. 529201) for 2 h at 40 °C. Afterwards, four amplification steps (AMP-1FL, 30 min; AMP-2FL, 15 min; AMP-3FL, 30 min; AMP-4FL, 15 min) were applied at 40 °C. Lastly, sections were covered with coverslips using fluorescent mounting medium (#S3023, Dako, CA, USA). Fluorescent images were acquired with a Carl Zeiss LSM 880 confocal microscope (Carl Zeiss, Oberkochen, Germany), tile scan was applied to obtain a whole view of the cingulate cortex.

### 4.5. Immunofluorescent Staining

Free-floating sections were washed in 1 × PBS for 5 min and subjected to antigen retrieval (Tris/EDTA, pH 9.0) for 10 min at 80 °C. After antigen retrieval, sections were washed in PBS for 5 min, followed by incubation in blocking buffer (5% normal donkey serum, 0.25% Triton-X in PBS) for 1 h at room temperature. Then, primary antibodies were applied, including rabbit anti-PSAP (1:100, #10801-1-AP, Proteintech, IL, USA), sheep anti-PGRN (1:100, #AF2557, R&D systems, Min, USA), mouse anti-Cre recombinase (1:100, MAB3120, Merck, NJ, USA). The following day, sections were washed in PBS for 5 min, three times, and incubated for 1 h at room temperature with corresponding fluorophore-conjugated secondary antibodies: donkey anti-mouse IgG Alexa Fluor 647 (1:500; A32787, Thermo Fisher Scientific, MA, USA), donkey anti-rabbit IgG Alexa Fluor 568 (1:500; A10042, Thermo Fisher Scientific), donkey anti-sheep IgG Alexa Fluor 405 (1:500; ab175676, Abcam, Cambs, UK). After washing in 1 × PBS, sections were mounted onto microscope slides and covered with coverslips using fluorescent mounting medium (S3023, Dako). Fluorescent images were obtained by using a Carl Zeiss LSM 880 confocal microscope. During the imaging process, settings were adjusted to minimize the background signal while obtain the best specific signal, avoiding saturation. The settings were kept the same for different groups of mouse brain sections. The mean fluorescence intensity was quantified by Image J (NIH, Bethesda, MD, USA).

### 4.6. Radioactive In Situ Hybridization

A ^35^S-labeled anti-sense cRNA probe against EGR-1 or ARC was prepared by in vitro transcription from a cDNA clone corresponding to fragments of EGR-1 or ARC, respectively. The transcription was performed with 100 ng of linearized plasmid using 35S-UTP (1000 Ci/mmol) and T3 RNA polymerase. In situ hybridization was performed as previously described (Zhang et al., 2007). Briefly, 12 μm thick cryostat sections were fixed in 4% PFA for 5 min at room temperature. After fixation, sections were rinsed twice in 4 × sodium chloride–sodium citrate buffer (SSC), followed by 10 min incubation in 0.25% acetic anhydride in 0.1 M triethanolamine/4 × SSC (pH 8) at room temperature. After dehydration in graded alcohols, the sections were hybridized in 50 μL of hybridization solution (20 mM Tris–HCl/1 mM EDTA/300 mM NaCl/50% formamide/10% dextran sulfate/1 × Denhardt’s/250 μg/mL yeast tRNA/100 μg/mL salmon sperm DNA/0.1% SDS/0.1% sodium thiosulphate) containing 35S-labeled EGR-1 or ARC probe, overnight, at 55 °C. The next day, slides were washed in 4 × SSC (5 min, four times), incubated in RNAse A (20 μg/mL) (20 min, at 37 °C), then sequentially washed in 2 × SSC (5 min, twice), 1 × SSC (5 min), 0.5 × SSC (5 min, twice) at room temperature, and rinsed in 0.1 × SSC at 65 °C (30 min, twice) (all washing buffer contained 1 mM DTT). After washing once in 0.1 × SSC at room temperature, the slides were dehydrated in graded ethanol. The slides were then exposed on X-ray films for 1–2 weeks. Autoradiograms were obtained with a Dia-Scanner (Epson Perfection 4870 PHOTO). Optical density values were measured in different regions using ImageJ, subtracting background signal.

### 4.7. Behavioral Tests

#### 4.7.1. Open Field Test (OFT)

The open field test was conducted in a 46 × 46 cm arena with a 30-lux indirect light. Mice were allowed to travel freely in the arena for 30 min and videotaped. A ceiling camera coupled to the EthoVision XT11.5 (Noldus, Gelderland, The Netherlands) software was used for video tracking. Distance traveled in the arena and time spent in the center area were measured automatically by the software.

#### 4.7.2. Light–Dark Transition Test (LDT)

The test was carried out in a box with dimensions of 45 × 30 × 30 cm, which is comprised of a small (one third) dark compartment and a large (two thirds) illuminated (200-lux indirect light) compartment. Mice were allowed to explore freely between two chambers for 15 min, which was videotaped by a ceiling camera coupled to the EthoVision XT11.5 (Noldus) software. Distance traveled and time spent in the light chamber were measured by the software.

#### 4.7.3. Prepulse Inhibition (PPI) Test

The test was performed in two startle chambers, each containing a Plexiglas cylinder attached to a platform and a loudspeaker that generated both a constant background noise (white noise 65 dB) and different acoustic stimuli. The startle response of the mice was recorded by a piezoelectric transducer under the platform. Before experiments, the loudspeaker and the sensitivity of the transducer platform of each startle chamber was calibrated. The test protocol starts with a 5 min background noise (65 dB) habituation period. Four blocks of trials were conducted following the habituation, with an average interval of 12 s. The first and final block contained five pulse-alone (40 ms 120 dB) trials. The second and third block have five trial types, including one pulse-alone trial, three prepulse-pulse trials (20 ms long prepulses (3 dB, 6 dB, 12 dB above background noise) presented before the onset of pulse-alone stimuli by a 100 ms interval), and one no-stimuli trial. Five trial types were pseudorandomized with five of each trial type in each block. Thus, the test contains 60 trials and lasts approximately 23 min. The pulse-alone reactivity (P) and prepulse-pulse reactivity (PP) were obtained in the pulse-alone trials and prepulse-pulse trials in the second and third block, as shown by the maximal response peak amplitude. The PPI is calculated according to the formula: %PPI = 100 × (P − PP)/P.

#### 4.7.4. Y Maze (YM)

The test was conducted in a light-grey colored polyvinylchloride symmetric Y-shaped compartment containing three equal-length arms (A, B, C), with dimensions of 40 × 8 × 20 cm. Mice were placed at the center of the Y-maze and allowed to explore freely for 8 min, which was videotaped by a ceiling camera. An alteration was counted when the mouse entered all three arms continuously. For instance, a pattern of A-B-C-B-A contains two alterations. Percentage of spontaneous alteration was calculated according to the formula: % Alteration = 100 × [alteration number/(total arm entries-2)].

#### 4.7.5. Forced Swim Test (FST)

The forced swim test was performed according to modified version of Porsolt’s protocol. Briefly, mice were placed in a vertical Plexiglas cylinder with a diameter of 20 cm and water depth of 15 cm. Temperature of the water was 23–25 °C. Mice were allowed to swim undisturbed in the cylinder for 6 min, which was videotaped, and dried before returning to their home cage. The last 4 min of the swimming was analyzed automatically by the EthoVision XT11.5 (Noldus) software. Mice were sacrificed 30 min after the FST for immediate early gene detection [55,56].

#### 4.7.6. Behavioral Tests

Timeline shown in Figure 5:

### 4.8. Statistics

All statistical analyses were performed with Prism 9.3.1 (GraphPad, CA, USA). Student’s *t*-test was applied for two-group comparisons. One-way analysis of variance (ANOVA) followed by Sidak’s post hoc test was used for multiple-group comparisons. Data are shown as mean ± S.E.M. Sample sizes are indicated in each figure legend. A *p*-value less than 0.05 was considered significant. * *p* < 0.05, ** *p* < 0.01, *** *p* < 0.001, **** *p* < 0.0001.

## Figures and Tables

**Figure 1 ijms-23-12056-f001:**
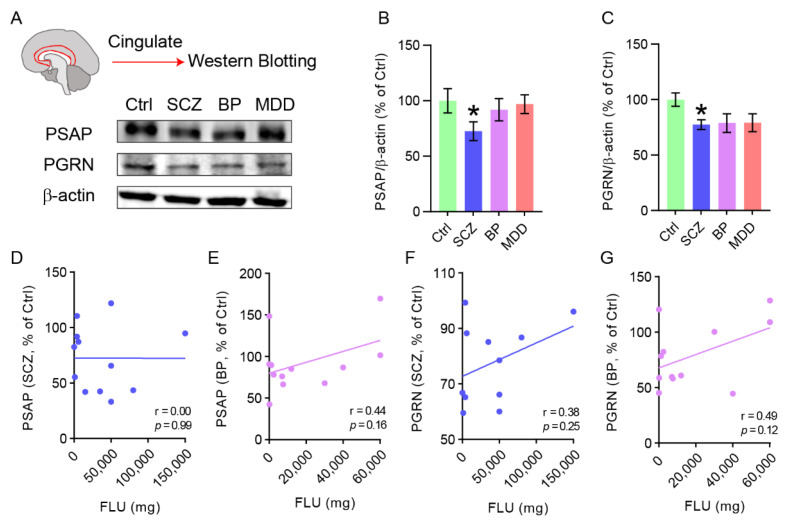
PSAP and PGRN levels are decreased in cingulate cortex tissue of schizophrenia patients. (**A**) Representative immunoblots of PSAP, PGRN, and β-actin in cingulate cortex tissue from subjects of healthy control (Ctrl), schizophrenia (SCZ), bipolar disorder (BP), and major depressive disorder (MDD). Upper panel, illustration of cingulate cortex in human brain. (**B**,**C**) Bar graphs presenting quantification of the ratio of protein levels of PSAP/β-actin (**B**) and PGRN/β-actin (**C**). (**D**,**E**) Scatter plots showing the Pearson’s correlation analysis of PSAP and lifetime quantity of fluphenazine or equivalent (FLU) in schizophrenia (**D**) and bipolar disorder group (**E**), respectively. Each dot represents one patient of SCZ (blue) or BP (magenta). (**F**,**G**) Scatter plots showing the Pearson’s correlation analysis of PGRN and FLU in schizophrenia (**F**) and bipolar disorder group (**G**), respectively. One-way ANOVA with Sidak’s post hoc test. *N* = 11–12 samples in each group. Data are presented as mean ± S.E.M. * *p* ˂ 0.05 compared to Ctrl.

**Figure 2 ijms-23-12056-f002:**
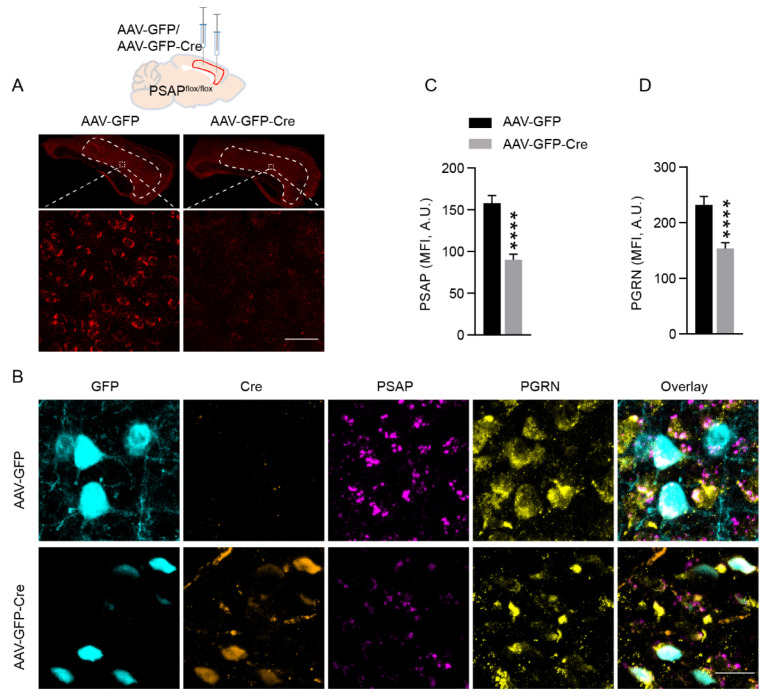
Conditional deletion of neuronal PSAP in the cingulate cortex causes downregulation of neuronal PGRN. (**A**) Representative fluorescent in situ hybridization images of PSAP mRNA in the cingulate cortex of mice injected with AAV-GFP and AAV-GFP-Cre. Upper row, schematic depiction of stereotaxic injection sites in one hemisphere (the same in the other hemisphere). Scale bar, 50 μm. (**B**) Representative immunofluorescent staining images of GFP (cyan), Cre (orange), PSAP (magenta), and PGRN (yellow) in the cingulate cortex of mice injected with AAV-GFP and AAV-GFP-Cre, respectively. Scale bar, 20 μm. (**C**,**D**) Bar graphs showing quantification of mean fluorescence intensity (MFI) of PSAP (**C**) and PGRN (**D**) staining in GFP- or Cre-positive cells. A.U, arbitrary unit. Student’s *t*-test, *N* = 34 cells in each group. Data are presented as mean ± S.E.M. **** *p* ˂ 0.0001.

**Figure 3 ijms-23-12056-f003:**
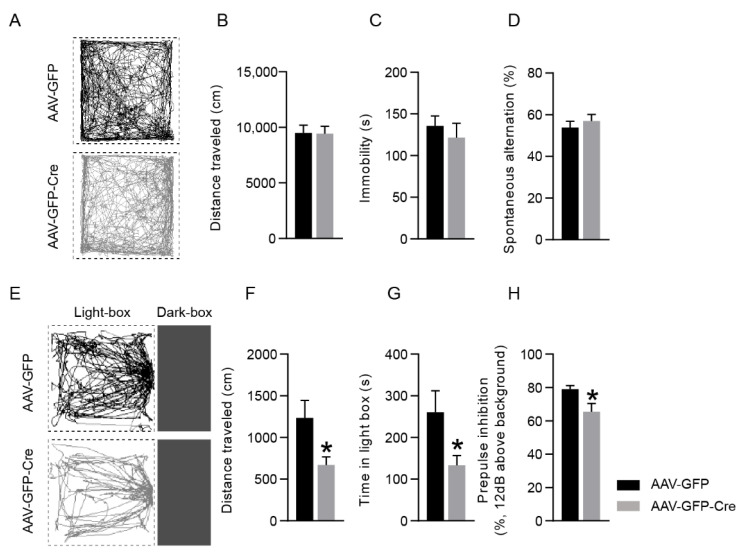
Behavioral characterization of AAV-injected mice. (**A**) Representative tracking images of mice in the open field test. (**B**) Quantification of distance traveled in the open field by two groups of mice. (**C**) Quantification of immobility time in the forced swim test. (**D**) Quantification of spontaneous alteration percentage in the Y-maze test. (**E**) Representative tracking images of mice in the light–dark transition test. (**F**,**G**) Quantification of distance traveled (**F**) and time spent (**G**) in the light box of two groups of mice. (**H**) Quantification of prepulse inhibition with a prepulse of 12 dB above background noise. Student’s *t*-test, *N* = 7–10 and 7–11 for AAV-GFP- and AAV-GFP-Cre-injected mice, respectively. Data are presented as mean ± S.E.M. * *p* ˂ 0.05.

**Figure 4 ijms-23-12056-f004:**
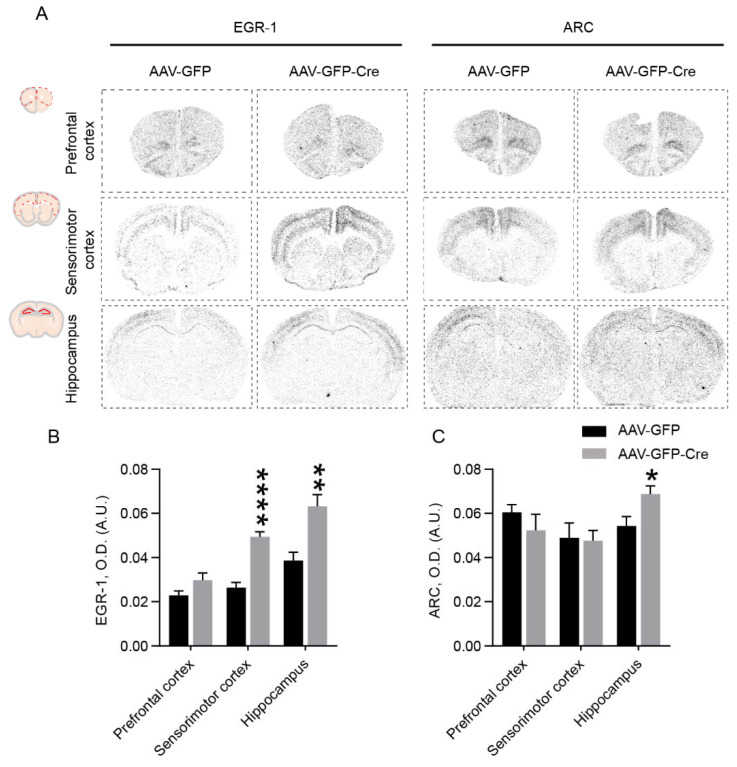
EGR-1 and ARC levels are elevated in different brain regions by neuronal PSAP deletion in the cingulate cortex. (**A**) Representative in situ hybridization autoradiographs of EGR-1 and ARC mRNA levels in different brain regions (prefrontal cortex, sensorimotor cortex, and hippocampus) of AAV-GFP- and AAV-GFP-Cre-injected mice. Upper left of each column shows quantification areas for each region. (**B**,**C**) Bar graphs showing quantification of optical density (O.D.) values of EGR-1 (**B**) and ARC (**C**) of the aforementioned brain regions. Student’s *t*-test, *N* = 6, 5 for AAV-GFP- and AAV-GFP-Cre-injected mice, respectively. Data are presented as mean ± S.E.M. * *p* ˂ 0.05, ** *p* ˂ 0.01, **** *p* ˂ 0.0001.

**Figure 5 ijms-23-12056-f005:**
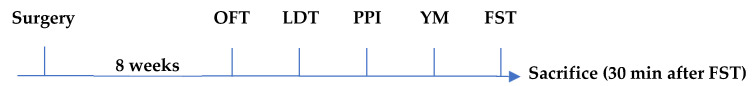
Timeline of behavioral tests.

**Table 1 ijms-23-12056-t001:** Demographics and clinical information of donors of anterior cingulate cortex tissue.

	Healthy Control	Schizophrenia	Bipolar Disorder	Major Depressive Disorder
**Age (y)**	49.4 ± 3.2	43.6 ± 4.2	40.0 ± 3.4	46.3 ± 2.4
**Gender (M/F)**	7/5	7/5	8/4	8/4
**Age of Onset (y)**	N/A	23.5 ± 2.6	20.9 ± 2.5	32.0 ± 3.3
**Psychosis (%)**	0	100	75	0
**FLU (mg)**	0	37083 ± 12738	18367 ± 6691	0
**PMI (H)**	23.6 ± 2.7	31.4 ± 4.0	32.1 ± 4.3	27.3 ± 3.4
**pH**	6.33 ± 0.06	6.19 ± 0.08	6.20 ± 0.07	6.20 ± 0.06

y, year; d, day; M, male; F, female; FLU, lifetime quantity of fluphenazine or equivalent; PMI, postmortem interval; H, hour; N/A, not applicable. Data are presented as mean ± S.E.M. There were no significant differences in age, gender, PMI, and pH values among the study groups according to one-way ANOVA.

## Data Availability

The datasets and materials used and/or analyzed during the current study are available from the corresponding author on reasonable request.

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
