# Peer review of "Decreased Prosaposin and Progranulin in the Cingulate Cortex Are Associated with Schizophrenia Pathophysiology"

_ijms, 2022, doi:10.3390/ijms231912056_

Round 1
Reviewer 1 Report
Recent evidence shows that PSAP and PGRN are vital players in neurodegenerative diseases. The authors provide an interesting study looking into the potential role of PSAP and PGRN in major mental disorders. The study provide more evidence for understanding the mechanism of schizophrenia. The paper is well written. Minor comments are:
1. The sample size in the expression analysis was relatively small, therefore, the authors may discuss the limitations of the study.
2. Since progranulin exerts inflammatory effects through NF-κB pathway, it is informative to mention recent evidence supporting the pathway in schizophrenia (PMID: 35524996, 33229121).
3. Please define the abbreviation PD.
Reviewer 2 Report
In this paper, the authors investigate the role of PSAP and PGRN in schizophrenia. They show reduced protein expression in post-mortem human tissue. Upon down regulation in mice, they show increased anxiety and reduced PPI. No changes were found in locomotor activity, spontaneous alternation and immobility in the forced swim test. Reduced PSAP in the cingulate cortex resulted in increased immediate early gene expression in the sensorimotor cortex and the hippocampus. The results are potentially interested as they reveal novel targets for schizophrenia. However, there are significant methodological details missing that are required to validate the results. Furthermore, the logic behind some experiments is not always clear.
- in the human samples, information on the quality of the samples (show similar protein density, information on sample handling post-mortem, information about the region of cingulate cortex used) is missing.
- In the animals, information on age, sex used is also missing. Furthermore, a timeline of the of the experiments conducted is also necessary. In particular, the IEG experiments, under what conditions were the brain isolated? After a behavioral task? Just from the home cage? This information is important to be able to interpret the results. Ideally, the IEG experiments should be conducted to reflect neuronal activation following PPI, as in the discussion the relationship between neuronal activity and PPI is discussed.
- The logic behind examining IEG expression is not clear. The authors mention the role of ERG-1 and ARC in schizophrenia, however the link between changes in lysosomal function and neuronal activity is not evident.
- How could the down regulation of PSAP increase the activity in sensorimotor cortex and hippocampus?
- While increased anxiety is a significant finding, it is not appropriately discussed.
- A discussion on the different cell types targeted by the viral construct is not mentioned. It seems that there is expression of both PSAP and PGRN in cells not targeted by the virus. There should be either experiments indicating the cell identity or discussion on existing knowledge about the presence of PSAP in both pyramidal neurons and interneurons.
- Relying on mean fluorescence intensity to claim reduction in protein levels is not very appropriate unless details on control conditions and measurements are mentioned. Stronger evidence would be provided via western blots.
Round 2
Reviewer 2 Report
The authors have answered my questions. However, some of their answers raised some other points.
The authors mentioned that they removed the brains 30min after FST. Why? Does 30min correspond to the timeline of Erg-1 and Arc expression? I was under the impression that Arc mRNA reaches maximum about 5-10 min following behavioral testing (PMID: 35149121 and 17446403), while for ERG-1 1hr is used for protein levels (PMID: 30195021). Appropriate papers that validate the 30min should be referenced.
Furthermore, since IEG expression is not related to PPI, I don’t see the discussion about PPI and neuronal activiation relevant. Instead, the authors should discuss more topics related to their findings.
Additionally, given the storage data of the human samples, it is evident that healthy samples were stored significantly less time than schizophrenia, bipolar or depression samples. A comment should also be made on this.
